# Association between chronic physical conditions and depressive symptoms among hospital workers in a national medical institution designated for COVID-19 in Japan

Ami Fukunaga[1]*, Yosuke Inoue[1], Shohei Yamamoto[1], Takako Miki[1], Dong Van Hoang[1], Rachana Manandhar Shrestha[1], Hironori Ishiwari[2], Masamichi Ishii[2], Kengo Miyo[2], Maki Konishi[1], Norio Ohmagari[3], Tetsuya Mizoue[1]

1 Department of Epidemiology and Prevention, Center for Clinical Sciences, National Center for Global Health and Medicine, Tokyo, Japan, 2 Center for Medical Informatics Intelligence, National Center for Global Health and Medicine, Tokyo, Japan, 3 Disease Control and Prevention Center, National Center for Global Health and Medicine, Tokyo, Japan

* afukunaga@hosp.ncgm.go.jp

**Data Availability Statement:** The data are not publicly available due to ethical restrictions for public deposition but available from the

## Abstract

### Objective

This study aimed to investigate the cross-sectional association between the presence of chronic physical conditions and depressive symptoms among hospital workers at a national medical institution designated for COVID-19 treatment in Tokyo, Japan. We also accounted for the combined association of chronic physical conditions and SARS-CoV-2 infection risk at work in relation to depressive symptoms, given that occupational infection risk might put additional psychological burden among those with chronic physical conditions with risk of severe COVID-19 outcome.

### Methods

The study sample consisted of 2,440 staff members who participated in a health survey conducted at the national medical institution during period between October 2020 and December 2020. Participants who reported at least one chronic physical condition that were deemed risk factors of severe COVID-19 outcome were regarded as having chronic physical conditions. Depressive symptoms were assessed using the patient health questionnaire-9 (PHQ-9). We performed logistic regression analysis to assess the association between chronic physical conditions and depressive symptoms.

### Results

Our results showed that the presence of chronic physical conditions was significantly associated with depressive symptoms (odds ratio (OR) = 1.49, 95% confidence interval (CI) = 1.10–2.02). In addition, the prevalence of depressive symptoms was significantly higher among healthcare workers with chronic physical conditions who were at a higher occupational infection risk (OR = 1.81, 95% CI = 1.04–3.16).

Department of Epidemiology and Prevention, Center for Clinical Sciences, National Center for Global Health and Medicine, Tokyo, Japan (yoboinfo@hosp.ncgm.go.jp) for researchers who meet the criteria to access the data.

**Funding:** The study was supported by NCGM COVID-19 Gift Fund and the Japan Health Research Promotion Bureau Research Fund (2020-B-09). The funders had no role in study design, data collection and analysis, decision to publish, or preparation of the manuscript.

**Competing interests:** The authors have declared that no competing interests exist.

## Conclusion

Our findings suggest the importance of providing more assistance to those with chronic physical conditions regarding the prevention and control of mental health issues, particularly among frontline healthcare workers engaging in COVID-19-related work.

## Introduction

Chronic physical conditions have been studied extensively in relation to mental health [1–3]. For example, the WHO World Health Survey reported a higher prevalence of depression in people with at least one chronic physical condition (9.3–23%) compared to those without any condition (3.2%) [3]. While chronic physical conditions and depression often coexist and their association can be bidirectional [4], one line of research has suggested that chronic physical conditions cause depression/depressive symptoms [1, 2] via physical symptoms (e.g., pain and functional impairment), decreased quality of life, fear of disease progression, burden of disease self-management, and medical costs [4].

Investigation on the link between chronic physical conditions and mental health is particularly important among healthcare workers during the current pandemic of coronavirus disease 2019 (COVID-19) given that chronic physical conditions, such as hypertension and diabetes, have been suggested as risk factors of severe COVID-19 symptoms or mortality [5, 6] and that they are at a substantial risk of the severe acute respiratory syndrome coronavirus 2 (SARS-CoV-2) infection and may perceive a higher vulnerability. More specifically, frontline health care workers with chronic physical conditions who are at higher occupational infection risk (e.g., having direct contact with infected patients) may be at a higher risk of depression as they are more susceptible to SARS-CoV-2 infection and adverse health outcomes as result of the infection [5, 6]. However, we are not aware of any study investigating the association between chronic physical conditions and mental health issues among healthcare workers. In addition, no study has investigated this subject with consideration of occupational infection risk.

The present study was designed to investigate the cross-sectional association between chronic physical conditions and depressive symptoms among hospital workers at a national medical institution designated for COVID-19 treatment in Japan. We also accounted for the combined association of the presence of chronic physical conditions and occupational infection risk in relation to depressive symptoms. We hypothesized that having chronic physical conditions might be associated with depressive symptoms, and that the magnitude of the association might be larger for individuals with chronic physical conditions who are at a higher occupational risk of SARS-CoV-2 infection.

## Methods

Data for the present study were derived from the Clinical Epidemiology Study on the SARS-CoV-2 antibody, an ongoing study conducted among workers at the National Center for Global Health and Medicine (NCGM) [7, 8]. The primary objective of this study was to investigate the prevalence and determinants of SARS-CoV-2 infection among workers. The first survey was conducted at one of the NCGM hospitals, located in Toyama, Tokyo, Japan in July 2020, which mainly targeted those who worked in COVID-19-related departments or were engaged in any COVID-19-related work. The second survey was conducted among a wider range of workers, along with health check-ups in October 2020 at the work site in Toyama,

Tokyo, and in December 2020 at another work site in Kohnodai, Chiba prefecture. For the present study, we used information from the second survey. Participants were asked to complete an electronic questionnaire on sociodemographic factors, lifestyle factors, COVID-19-related work factors, chronic physical conditions, and depressive symptoms.

A total of 2,893 people were invited to participate in the survey, and 2,480 agreed to participate (response rate: 85.7%). For the present study, we excluded those with missing information on the exposure, outcome, and covariates (described below) (n = 35). In addition, we excluded those who reported having depression (n = 5) as it is possible that they experienced the outcome before the onset of COVID-19 pandemic. After applying the exclusion criteria, the sample consisted of 2,440 participants aged 18–75 years (1,698 women and 742 men). Written informed consent was obtained from all the participants. The study protocol was approved by the NCGM ethics committee (approval number: NCGM-G-003598).

## Exposure

In this study, we defined the presence of chronic physical conditions using information on chronic physical conditions that might be risk factors for severe COVID-19 outcome [5, 6]. We asked participants whether they have any of the following chronic physical conditions: diabetes, hypertension, chronic obstructive pulmonary disease (COPD) or bronchial asthma, cardiovascular disease, cerebrovascular disease, cancer, or other chronic physical conditions. The response options for each condition included "no," "yes (under medication)," and "yes (untreated)" (plus "yes [diet therapy]" for diabetes). For other chronic physical conditions, participants were asked to specify the conditions. From the list of other conditions, we picked out the following chronic physical conditions that meet our definition of exposure: dyslipidemia [9], thyroid disorders [10], anemia [11], human immunodeficiency virus (HIV) infection [6], neurological disorders [12], and autoimmune diseases [13].

For additional analysis, we included obesity [6] as a chronic physical condition. In the present study, height and weight were self-reported via the electronic questionnaire, and body mass index (BMI) was calculated by dividing the weight (kg) by the square of the height ($m^2$). Obesity was defined as a BMI of $\geq$30 kg/$m^2$.

## Assessment of depressive symptoms

The Japanese version of the Patient Health Questionnaire 9 (PHQ-9) [14] was used to assess depressive symptoms. The PHQ-9 consists of nine items, with each item scored on a scale of 0–3 based on the frequency of depressive symptoms in the past two weeks (0 = not at all, 1 = several days, 2 = more than half of the days, and 3 = nearly every day). The total score ranged from 0 to 27, with higher scores indicating a greater severity of depressive symptoms. Participants with a PHQ-9 score of $\geq$10 were regarded as having depressive symptoms, which has been validated for its assessment [14, 15].

## Other variables

We obtained information on age (continuous), sex (male or female), and occupation from the labor management office at the NCGM. Information on other variables, including living arrangements, smoking status, alcohol consumption, physical activity, diet, sleep duration, working hours, and COVID-19-related factors was obtained via the electronic questionnaire.

We grouped occupations into five categories: doctors, nurses, allied healthcare professionals, administrative staff, and others. Information on the number of cohabitants (living alone, living with one person, living with two, three, four, or five or more people) was used to define living arrangements (living alone or living with others). For current smoking status (yes or

no), we defined smokers as those who smoked cigarettes and/or heat-not-burn cigarettes and non-smokers as those who did not smoke cigarettes and heat-not-burn cigarettes. Daily alcohol consumption was categorized into four groups: none, <1, 1–<2, or ≥2 go/day (go: a Japanese traditional unit [180 mL]). Weekly leisure-time physical activity was categorized into five groups: none, <60, 60–120, 120–180, or ≥180 minutes/week. We obtained information on the frequency of balanced-meal consumption by asking "How frequently did you have meals with a combination of staple foods (e.g., rice, bread, noodles), main dishes (dishes made of meat, fish, eggs, soy products, etc.), and side dishes (small bowls or small dishes made of vegetables, mushrooms, potatoes, seaweed, etc.)?", with the following response options: rarely, 2–3 days/week, 4–5 days/week, or almost every day. Sleep duration was categorized into three groups: <6, 6–7, or ≥7 hours. BMI was categorized into five groups: <18.5, 18.5–<23, 23–<25, 25–<30, or ≥30 kg/m$^2$.

Self-reported information on working hours was categorized into three groups: ≤8, 9–10, or ≥11 hours/day. In relation to engagement in COVID-19-related work, we asked the following two questions: "Have you ever engaged in COVID-19-related work?" (yes or no) and "Did you engage in any work in which you were heavily exposed to SARS- CoV-2?" (yes or no). Using these two questions, we defined the degree of possible exposure to SARS-CoV-2 at work, which was categorized into three groups: low (those who did not engage in COVID-19-related work), moderate (those who engaged in COVID-19-related work without high exposure to SARS-CoV-2 at work), and high (those who engaged in COVID-19-related work with high exposure to SARS-CoV-2 at work). Regarding participants' experiences in relation to COVID-19, we asked them if they agreed with the following two statements: "You and your family have been bad-mouthed" (yes or no) and "I felt that I was discriminated against in some way" (yes or no). If participants answered "yes" to either question, they were considered as having experiences of being bad-mouthed or discriminated against in relation to COVID-19. We incorporated the discrimination experience in the present study because such experiences have been suggested to worsen mental health [16].

## Statistical analysis

Multiple logistic regression analysis was performed to investigate the association between chronic physical conditions and depressive symptoms. Model 1 was adjusted for age, sex, worksite, and occupation. Model 2 was additionally adjusted for living arrangements, smoking status, alcohol consumption, leisure-time physical activity, balanced-meal consumption, sleep duration, working hours, degree of possible exposure to SARS-CoV-2 at work, and being bad-mouthed or discriminated against in relation to COVID-19.

To examine the association while taking possible occupational infection risk into account, we established a variable by combining information about the presence of chronic physical conditions (yes or no) and degree of possible exposure to SARS-CoV-2 at work (low, moderate, or high).

All the statistical analyses were conducted using SAS version 9.4 (SAS Institute). Statistical significance was set at a $p$-value of <0.05 (two-tailed).

## Results

In the present study, 359 (14.7%) had depressive symptoms out of 2,440 participants. Table 1 presents the characteristics of the participants. The proportions of doctors, nurses, allied healthcare professionals, administrative staff, and those with other occupations were 16.8%, 36.6%, 14.9%, 11.6%, and 20.0%, respectively. A total of 1,275 participants (52.3%) had ever engaged in COVID-19-related work. Out of them, 582 (23.9%) had ever engaged in COVID-

**Table 1. Characteristics of study participants (n = 2,440).**

| | All (n = 2,440) | Chronic physical condition(s) | |
|---|---|---|---|
| | | No (n = 1,954) | Yes (n = 486) |
| Age, mean [SD] | 38.7 [12.0] | 36.7 [11.0] | 46.9 [12.4] |
| Sex (female), n (%) | 1,698 (69.6) | 1,399 (71.6) | 299 (61.5) |
| Work site | | | |
| Toyama (Tokyo) | 1,951 (80.0) | 1,592 (81.5) | 359 (73.9) |
| Kohnodai (Chiba) | 489 (20.0) | 362 (18.5) | 127 (26.1) |
| Occupation, n (%) | | | |
| Doctors | 410 (16.8) | 334 (17.1) | 76 (15.6) |
| Nurses | 894 (36.6) | 760 (38.9) | 134 (27.6) |
| Allied healthcare professionals | 364 (14.9) | 301 (15.4) | 63 (13.0) |
| Administrative staff | 284 (11.6) | 210 (10.8) | 74 (15.2) |
| Others | 488 (20.0) | 349 (17.9) | 139 (28.6) |
| Working hours, n (%) | | | |
| ≤8 hours/day | 1,208 (49.5) | 937 (48.0) | 271 (55.8) |
| 9–10 hours/day | 918 (37.6) | 772 (39.5) | 146 (30.0) |
| ≥11 hours/day | 314 (12.9) | 245 (12.5) | 69 (14.2) |
| Ever engaged in COVID-19-related work (yes), n (%) | 1,275 (52.3) | 1,038 (53.1) | 237 (48.8) |
| Degree of possible exposure to SARS-CoV-2 at work, n (%) | | | |
| Low | 1,165 (47.8) | 916 (46.9) | 249 (51.2) |
| Moderate | 693 (28.4) | 558 (28.6) | 135 (27.8) |
| High | 582 (23.9) | 480 (24.6) | 102 (21.0) |
| Being bad mouthed or discriminated against in relation to COVID-19 (yes), n (%) | 235 (9.6) | 173 (8.9) | 62 (12.8) |
| Living arrangements (living alone), n (%) | 907 (37.2) | 795 (40.7) | 112 (23.1) |
| Current smoking (yes), n (%) | 170 (7.0) | 139 (7.1) | 31 (6.4) |
| Alcohol drinking, n (%) | | | |
| None | 828 (33.9) | 653 (33.4) | 175 (36.0) |
| <1 go/day | 1,328 (54.4) | 1,094 (56.0) | 234 (48.2) |
| 1–<2 go/day | 211 (8.7) | 152 (7.8) | 59 (12.1) |
| ≥2 go/day | 73 (3.0) | 55 (2.8) | 18 (3.7) |
| Leisure time physical activity, n (%) | | | |
| None | 579 (23.7) | 463 (23.7) | 116 (23.9) |
| <60 minutes/week | 1,069 (43.8) | 858 (43.9) | 221 (43.4) |
| 60–<120 minutes/week | 444 (18.2) | 367 (18.8) | 77 (15.8) |
| 120–<180 minutes/week | 175 (7.2) | 135 (6.9) | 40 (8.2) |
| ≥180 minutes/week | 173 (7.1) | 131 (6.7) | 42 (8.6) |
| Balanced-meal consumption, n (%) | | | |
| Rarely | 431 (17.7) | 368 (18.8) | 63 (13.0) |
| 2–3 days/week | 714 (29.3) | 592 (30.3) | 122 (25.1) |
| 4–5 days/week | 482 (19.8) | 377 (19.3) | 105 (21.6) |
| Almost every day | 813 (33.3) | 617 (31.6) | 196 (40.3) |
| Sleep duration, n (%) | | | |
| <6 hours | 1,230 (50.4) | 947 (48.5) | 283 (58.2) |
| 6–<7 hours | 864 (35.4) | 716 (36.6) | 148 (30.5) |
| ≥7 hours | 346 (14.2) | 291 (14.9) | 55 (11.3) |
| BMI (kg/m$^2$), n (%) | | | |
| <18.5 | 278 (11.4) | 246 (12.6) | 32 (6.6) |
| 18.5–<23 | 1,461 (59.9) | 1,234 (63.2) | 227 (46.7) |

*(Continued)*

**Table 1.** (Continued)

| | All (n = 2,440) | Chronic physical condition(s) | |
| --- | --- | --- | --- |
| | | No (n = 1,954) | Yes (n = 486) |
| 23–<25 | 336 (13.8) | 253 (13.0) | 83 (17.1) |
| 25–<30 | 299 (12.3) | 188 (9.6) | 111 (22.8) |
| ≥30 | 66 (2.7) | 33 (1.7) | 33 (6.8) |

19-related work with high exposure to SARS-CoV-2. Among 2,440 participants, 486 (19.9%) had one or more chronic physical conditions. Among these participants, the cases of hypertension, COPD or bronchial asthma, diabetes, cardiovascular diseases, cancers, cerebrovascular diseases, and other chronic physical conditions were 166, 95, 55, 38, 29, 16, and 246, respectively (110 participants had more than one conditions) (S1 Table).

Table 2 shows the results of the multiple logistic regression analysis investigating the association between chronic physical conditions and depressive symptoms. Compared with those without any chronic physical condition, the odds ratio (OR) (95% confidence interval [CI]) of depressive symptoms was 1.49 (95% CI = 1.10–2.02) for those with chronic physical condition(s) in the fully adjusted model. The association remained the same when obesity was included as a chronic physical condition (OR = 1.41, 95% CI = 1.03–1.92).

The OR of depressive symptoms was significantly higher among those with chronic physical condition(s) who had high exposure to SARS-CoV-2 at work (i.e., high occupational infection risk) (OR = 1.81, 95% CI = 1.04–3.16) than among those without any chronic physical condition who did not engage in COVID-19-related work (i.e., low occupational infection risk) (Table 3), though we did not find any evidence of a significant interaction between the presence of chronic physical condition(s) and occupational infection risk (p for interaction = 0.23). The combined association of chronic physical condition(s) and SARS-CoV-2 infection risk at work with depressive symptoms is shown in Fig 1.

## Discussion

In the present study, we found that the presence of chronic physical condition(s) was significantly associated with depressive symptoms among workers at a national medical institution

**Table 2. Odds ratios and 95% confidence intervals of depressive symptoms according to the presence of chronic physical condition(s).**

| | Cases/Subjects | Model 1 | Model 2 |
| --- | --- | --- | --- |
| **Chronic physical condition(s)** | | | |
| No | 274/1,954 | 1.00 (reference) | 1.00 (reference) |
| Yes | 85/486 | **1.60 (1.20–2.14)** | **1.49 (1.10–2.02)** |
| **Chronic physical condition(s) (including obesity)*** | | | |
| No | 271/1,921 | 1.00 (reference) | 1.00 (reference) |
| Yes | 88/519 | **1.49 (1.12–1.98)** | **1.41 (1.03–1.92)** |

Model 1 is adjusted by age (years, continuous), sex (male or female), work site (Toyama or Kohnodai), and occupation (doctors, nurses, allied health care professionals, administrative staff, or others).

Model 2 is adjusted by variables in model 1, and living arrangements (living alone or living with others), current smoking (yes or no), alcohol consumption (none, <1, 1–<2, or ≥2 go/day), leisure time physical activity (none, <60, 60–<120, 120–<180, or ≥180 minutes/week), balanced-meal consumption (rarely, 2–3, 4–5 days/week, or almost every day), sleep duration (<6, 6–<7, or ≥7 hours), BMI (<18.5, 18.5–<23, 23–<25, 25–<30, or ≥30 kg/m$^2$), working hours (≤8, 9–10, or ≥11 hours/day), degree of possible exposure to SARS-CoV-2 at work (low, moderate, or high), and being bad mouthed or discriminated against in relation to COVID-19 (yes or no).

*Obesity (BMI ≥30 kg/m$^2$) is considered as one of chronic physical conditions so that BMI is not adjusted in models.

**Table 3. Odds ratios and 95% confidence intervals of depressive symptoms according to the combination of the presence of chronic physical condition(s) and degree of possible exposure to SARS-CoV-2 at work.**

| Degree of possible exposure to SARS-CoV-2 at work | Chronic physical condition(s) | | | | | |
|---|---|---|---|---|---|---|
| | No (n = 1954) | | | Yes (n = 486) | | |
| | Low* (n = 916) | Moderate† (n = 558) | High‡ (n = 480) | Low* (n = 249) | Moderate† (n = 135) | High‡ (n = 102) |
| Cases | 132 | 69 | 73 | 38 | 25 | 22 |
| Model 1 | 1.00 (ref) | 0.83 (0.60–1.14) | 1.13 (0.81–1.58) | 1.31 (0.87–1.97) | **1.70 (1.04–2.77)** | **2.00 (1.18–3.41)** |
| Model 2 | 1.00 (ref) | 0.75 (0.54–1.05) | 1.03 (0.73–1.46) | 1.23 (0.80–1.87) | 1.37 (0.82–2.28) | **1.81 (1.04–3.16)** |

Model 1 is adjusted by age (years, continuous), sex (male or female), work site (Toyama or Kohnodai), and occupation (doctors, nurses, allied health care professionals, administrative staff, or others).

Model 2 is adjusted by variables in model 1, and living arrangements (living alone or living with others), current smoking (yes or no), alcohol consumption (none, <1, 1–<2, or ≥2 go/day), leisure time physical activity (none, <60, 60–<120, 120–<180, or ≥180 minutes/week), balanced-meal consumption (rarely, 2–3, 4–5 days/week, or almost every day), sleep duration (<6, 6–<7, or ≥7 hours), BMI (<18.5, 18.5–<23, 23–<25, 25–<30, or ≥30 kg/m$^2$), working hours (≤8, 9–10, or ≥11 hours/day), and being bad mouthed or discriminated against in relation to COVID-19 (yes or no).

*Low: those who did not engage in COVID-19-related work.

†Moderate: those who engaged in COVID-19-related work without high exposure to SARS-CoV-2 at work.

‡High: those who engaged in COVID-19-related work with high exposure to SARS-CoV-2 at work.

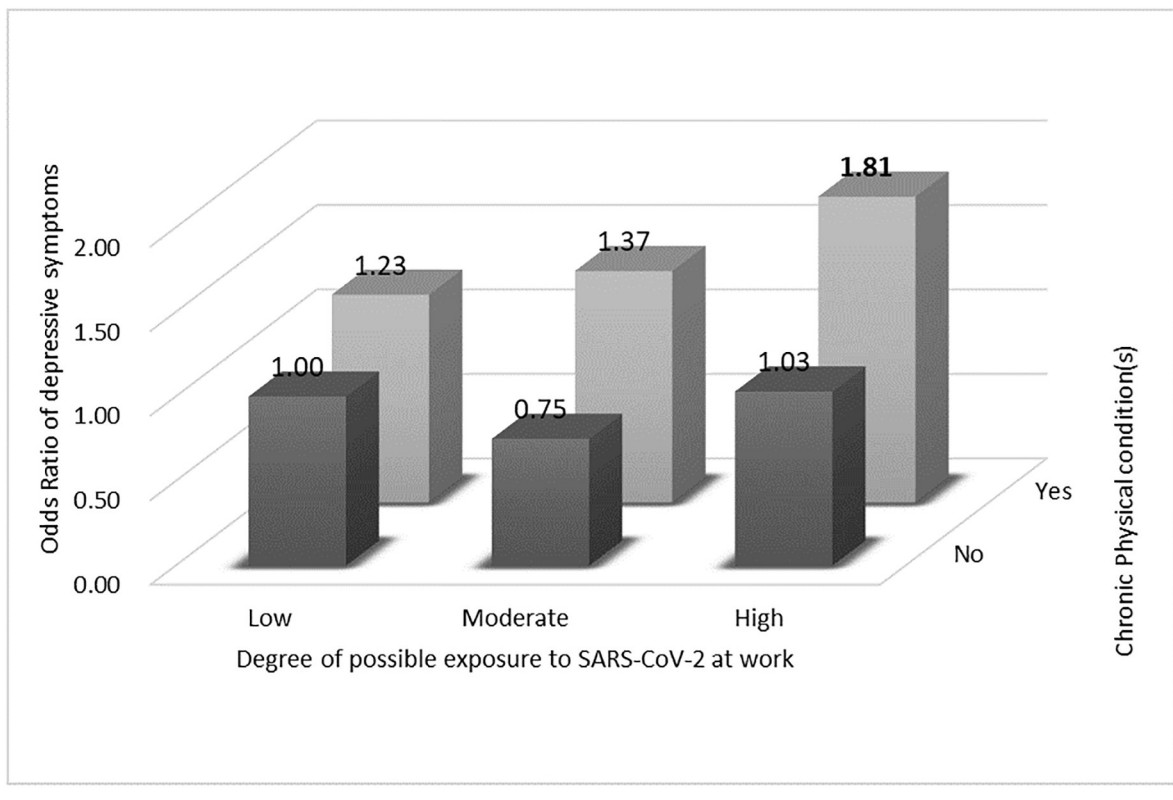

**Fig 1. The combined association of chronic physical condition(s) and degree of possible exposure to SARS-CoV-2 at work with depressive symptoms.**

designated for COVID-19 treatment in Japan. The magnitude of the association was larger in healthcare workers who had a higher occupational risk of SARS-CoV-2 infection than those who did not engage in COVID-19-related work.

While we are not aware of any study that examined the association between chronic physical conditions and depression/depressive symptoms among healthcare workers during this pandemic, our findings were consistent with those reported in previous studies that examined the association among general populations [17, 18]. For example, a recent systematic review [17] of three cross-sectional studies [19–21] concluded that individuals with chronic physical conditions were more likely to have mental illness (e.g., depressive symptoms, anxiety symptoms) than those without any chronic physical condition during the COVID-19 pandemic.

We also found that the magnitude of the association was greater for healthcare workers with chronic physical condition(s) who engaged in COVID-19-related work with a potential risk of SARS-CoV-2 infection compared with those who did not engage in COVID-19-related work. A possible interpretation is that those with such conditions who were at a higher occupational risk of SARS-CoV-2 infection may have higher perceived psychological stress, given that the combination of these conditions may put them at a higher risk of becoming severely ill or dying due to COVID-19 [5, 6]. For example, recent studies revealed that working at the frontlines and having direct contact with infected patients contributed to a higher proportion of mental health issues among healthcare workers during this pandemic [22–24]. Psychological stress associated with such working conditions (e.g., high degree of exposure to the virus and fear of infection) [25, 26] may worsen mental health issues in those with chronic physical conditions. Thus, frontline healthcare workers, especially those with chronic physical conditions, are physically and psychologically challenged while committing themselves to caring for infected patients, given that they are at a high risk of infection and are more susceptible to adverse health outcomes.

The present as well as previous studies conducted among the general population [17, 18] provide robust evidence on the association between chronic physical conditions and depression/depressive symptoms during the COVID-19 pandemic. Given that there has been a growing number of ageing population and thus, people with chronic physical conditions in Japan [27] as well as other countries [28], the association should be more extensively studied. It is possible that people with chronic physical conditions experience depressive symptoms as they might have fear of being infected and developing severe COVID-19 outcome or they might isolate themselves and make adverse changes in lifestyle behaviors (e.g., decreased physical activity, alcohol drinking) as a consequence of perceiving the risk and stress associated with having chronic physical conditions, which may worsen their physical conditions and affect mental health. Such psychological stress might further exacerbate their physical conditions and affect mental health. Thus, this finding underscores the urgency that we should be aware of their conditions and pay further attention for both physical and psychological care management and treatment for people with chronic physical conditions.

The major strength of the present study was taking the occupational infection risk into account while examining the association between chronic physical conditions and depressive symptoms. However, this study had several limitations. First, the cross-sectional design did not allow assessment of the temporal association between chronic physical conditions and depressive symptoms. Second, the information used in this study was self-reported, which might have been subject to recall bias. Third, we did not have detailed information on the severity of each chronic physical condition, which might have been an important omission for investigating the association with depressive symptoms. Fourth, we assessed depressive symptoms using a self-administered questionnaire without clinical diagnosis by a psychiatrist; however, the PHQ-9 has been validated to assess depressive symptoms. Finally, this study was

conducted in a medical institution designated for COVID-19; thus, the findings might not be generalizable to other settings.

In conclusion, this cross-sectional study provided evidence on the association between chronic physical conditions and depressive symptoms during the COVID-19 pandemic. Our findings warrant paying more attention and providing more assistance to those with chronic physical conditions regarding the prevention and control of mental health issues, particularly among frontline healthcare workers engaging in COVID-19-related work.

## Supporting information

**S1 Table. Information on chronic physical conditions (n = 486)**$^*$**.**
(DOCX)

## Acknowledgments

We thank study participants and staff members of NCGM and Haruka Osawa (NCGM) for administrative support.

## Author Contributions

**Conceptualization:** Ami Fukunaga, Yosuke Inoue, Shohei Yamamoto, Takako Miki, Dong Van Hoang, Rachana Manandhar Shrestha, Hironori Ishiwari, Masamichi Ishii, Kengo Miyo, Maki Konishi, Norio Ohmagari, Tetsuya Mizoue.

**Data curation:** Shohei Yamamoto, Maki Konishi, Tetsuya Mizoue.

**Formal analysis:** Ami Fukunaga.

**Investigation:** Ami Fukunaga, Yosuke Inoue, Shohei Yamamoto, Takako Miki, Dong Van Hoang, Rachana Manandhar Shrestha, Hironori Ishiwari, Masamichi Ishii, Kengo Miyo, Maki Konishi, Norio Ohmagari, Tetsuya Mizoue.

**Supervision:** Tetsuya Mizoue.

**Visualization:** Ami Fukunaga.

**Writing – original draft:** Ami Fukunaga.

**Writing – review & editing:** Ami Fukunaga, Yosuke Inoue, Shohei Yamamoto, Takako Miki, Dong Van Hoang, Rachana Manandhar Shrestha, Hironori Ishiwari, Masamichi Ishii, Kengo Miyo, Maki Konishi, Norio Ohmagari, Tetsuya Mizoue.

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
