## [Decision Letter · Decision Letter 0]

24 Jan 2022

PONE-D-21-28081Association between chronic physical conditions and depressive symptoms among hospital workers in a national medical institution designated for COVID-19 in JapanPLOS ONE

Dear Dr. Fukanaga,

Thank you for submitting your manuscript to PLOS ONE. After careful consideration, we feel that it has merit but does not fully meet PLOS ONE’s publication criteria as it currently stands. Therefore, we invite you to submit a revised version of the manuscript that addresses the points raised during the review process.

We look forward to receiving your revised manuscript.

Kind regards,

Stephan Doering, M.D.

Academic Editor

PLOS ONE

https://journals.plos.org/plosone/s/file?id=ba62/PLOSOne_formatting_sample_title_authors_affiliations.pdf”.

“The study was supported by NCGM COVID-19 Gift Fund and the Japan Health Research Promotion Bureau Research Fund (2020-B-09).”

6. Please include a separate caption for each figure in your manuscript.

Reviewers' comments:

Reviewer's Responses to Questions

**Comments to the Author**

1. Is the manuscript technically sound, and do the data support the conclusions?

Reviewer #1: Yes

Reviewer #2: Partly

2. Has the statistical analysis been performed appropriately and rigorously? 

Reviewer #1: Yes

Reviewer #2: Yes

3. Have the authors made all data underlying the findings in their manuscript fully available?

Reviewer #1: Yes

Reviewer #2: No

4. Is the manuscript presented in an intelligible fashion and written in standard English?

Reviewer #1: Yes

Reviewer #2: Yes

5. Review Comments to the Author

Reviewer #1: This study of chronic physical conditions and depression among over 2000 healthcare workers at a COVID-19 treatment center in Japan provides important insight into the drivers of mental health issues during COVID-19 for this population. The methods are strong and the manuscript is well written.

METHODS

Ln 42. It’s not clear why participants who reported depression, hay fever, or low back pain were excluded, is it just because these were not the chronic physical conditions of interest? If so, I would frame in terms of inclusion rather than exclusion criteria.

Ln 99-101. Please provide more context for the inclusion of these questions about being bad-mouthed or discriminated against. Is this because they work in a hospital designated for COVID-19 treatment? t’s not clear how this is COVID-related or how it’s relevant to the research question. If there is a culturally specific rationale, it would be helpful to explain this.

DISCUSSION

It would be good to place these findings in context with the physical and mental health of the Japanese population more generally.

Also, how is mental health conceptualized in Japan more broadly? Is there a lot of stigma that could result in underreporting, particularly in the healthcare worker population?

Reviewer #2: Thanks for the opportunity to review the manuscript "Association between chronic physical conditions and depressive symptoms among hospital workers in a national medical institution designated for COVID-19 in Japan". The work deals with an interesting topic. However, I think some adjustments are needed before publication. I hope you find my comments useful.

I would rewrite the abstract to provide a more concise and detailed idea of your work. Specify how the influence of the risk of SARS-CoV-2 infection has been considered. Try replacing "workers with the condition" with something clearer. Also, I would write from October to December 2020 instead of “or”.

The introduction section should be recasted. The authors need to provide a theoretical background on the relationship between chronic physical conditions and depressive symptoms, which theories and theoretical frameworks can justify the association and explain the underlying mechanisms (e.g., loss of autonomy, social isolation, redefinition of one's identity etc.), previous studies on the topic trying to focus on some of the specific conditions of the sample. Then, try to contextualize everything with reference to the COVID-19 emergency, I think this aspect should be emphasized also to provide a novelty. If you hypothesize that the risk of SARS-CoV-2 infection has an influence on the relationship between physical and depressive symptoms, try to justify why.

Line 16. You should put the reference about the scarcity of the studies.

In the methods sections you write chronic diseases, I think this definition (instead of physical symptoms) should be used even in the abstract/introduction and throughout the paper.

Try to explain more clearly how you classified the groups based on the degree of possible exposure to SARS-CoV-2, I understood the process but I think the authors could rewrite it to explain it better.

I suggest modifying the discussion section to better explain the findings by describing how they fit into the context of the previous literature and directions for future research. Consider adding some clinical and practical implications.

6. PLOS authors have the option to publish the peer review history of their article (what does this mean?). If published, this will include your full peer review and any attached files.

Reviewer #1: No

Reviewer #2: No

---

## [Author Response · Author response to Decision Letter 0]

16 Mar 2022

We uploaded the response letter for the editor and reviewers. Please see the attached document.

---

## [Editor Report · Decision Letter 1]

18 Mar 2022

Association between chronic physical conditions and depressive symptoms among hospital workers in a national medical institution designated for COVID-19 in Japan

PONE-D-21-28081R1

Dear Dr. Fukunaga,

We’re pleased to inform you that your manuscript has been judged scientifically suitable for publication and will be formally accepted for publication once it meets all outstanding technical requirements.

Kind regards,

Stephan Doering, M.D.

Academic Editor

PLOS ONE

---

## [Editor Report · Acceptance letter]

30 Mar 2022

PONE-D-21-28081R1 

Association between chronic physical conditions and depressive symptoms among hospital workers in a national medical institution designated for COVID-19 in Japan 

Dear Dr. Fukunaga:

I'm pleased to inform you that your manuscript has been deemed suitable for publication in PLOS ONE. Congratulations! Your manuscript is now with our production department. 

Kind regards, 

on behalf of

Professor Stephan Doering 

Academic Editor

PLOS ONE